# Establishment of two assays based on reverse transcription recombinase-aided amplification technology for rapid detection of H5 subtype avian influenza virus

Yang Li,[1] Jiajing Shang,[1,2] Yixin Wang,[3] Juan Luo,[1,2] Wenming Jiang,[1] Xin Yin,[1] Fuyou Zhang,[1] Chunran Deng,[1,2] Xiaohui Yu,[1] HuaLei Liu[1]

**ABSTRACT**  The avian influenza virus (AIV), a member of the type A influenza virus group, is harbored by avian hosts. The highly pathogenic H5 subtype of AIV has inflicted significant financial damage on the poultry industry and also presents a possible pandemic hazard to the worldwide human population. There is an urgent need for a detection method that is both rapid and accurate in identifying H5-AIV. In this study, two assays based on reverse transcription recombinase-aided amplification technology (RT-RAA) have been developed to detect H5-AIV, namely, real-time fluorescence and reverse transcription recombinase-aided amplification (RF-RT-RAA) and reverse transcription recombinase-aided amplification combined lateral flow dipstick (RT-RAA-LFD). The completion of the two methods can be achieved in 30 min, and there was no cross-reaction with the nucleic acid of other avian pathogens. The results indicated that the lowest detectable limit of RF-RT-RAA and RT-RAA-LFD for H5-AIV detection was 1 copy/µL, which was 100 times higher than that of RT-PCR ($10^2$ copies/µL). A total of 376 samples, consisting of 350 clinical and 26 experimental samples, underwent testing utilizing virus isolation, RF-RT-RAA and RT-RAA-LFD methods, respectively. The results demonstrated a high degree of consistency between those assays. In summary, both the RF-RT-RAA and RT-RAA-LFD methods demonstrated excellent specificity and sensitivity, making them ideal for use in both laboratory and field settings.

**IMPORTANCE**  Avian influenza virus (AIV) subtype H5 is a highly contagious zoonotic disease and a serious threat to the farming industry and public health. Traditional detection methods, including virus isolation and real-time PCR, require tertiary biological laboratories and are time-consuming and complex to perform, making it difficult to rapidly diagnose H5 subtype avian influenza viruses. In this study, we successfully developed two methods, namely, RF-RT-RAA and RT-RAA-LFD, for rapid detection of H5-AIV. The assays are characterized by their high specificity, sensitivity, and user-friendliness. Moreover, the results of the reaction can be visually assessed, which are suitable for both laboratory testing and grassroots farm screening for H5-AIV.

**KEYWORDS**  H5-AIV, RF-RT-RAA, RT-RAA-LFD, rapid detection

Address correspondence to HuaLei Liu, liuhualei@cahec.cn.

Yang Li, Jiajing Shang, and Yixin Wang contributed equally to this article. Author order was determined on the basis of contributions.

The authors declare no conflict of interest.

See the funding table on p. 12.

Avian influenza is caused by avian influenza virus (AIV), a member of the *Orthomyxoviridae* family, which brought serious threats to both animal and human health. AIV can infect a multitude of animal species, including but not limited to chickens, turkeys, pigs, and humans, causing high mortality rates in infected birds and even leading to significant economic losses for farmers (1). Throughout 2022, the AIV outbreak has persisted worldwide, resulting in a significant loss of avian life. According to data from the World Organization for Animal Health (WOAH) (https://www.woah.org/en), over

138 million birds have perished globally between September 2021 and September 2022, surpassing the combined mortality rate of the past 5 years.

The classification of AIV was established based on the level of pathogenicity as either highly pathogenic avian influenza or low-pathogenic avian influenza (LPAI) (2). Poultry-origin strains that possess the hemagglutination (HA) gene of the H5 or H7 subtypes have exhibited a high degree of pathogenicity, leading to considerable challenges for the global poultry industry. AIV of the H5 subtype (H5-AIV) is known to cause high mortality rates in both domestic poultry and wild birds worldwide (3). Its zoonotic potential is a matter of concern for public health. In 1996, the highly pathogenic H5N1 AIV was initially identified in a group of diseased geese in Guangdong, China, and was designated as A/Goose/Guangdong/1/96, GS/GD/96 (4). The virus initially emerged in China, but it eventually spread to the Middle East and Europe. Additionally, there were reports of the same virus in wild birds in various places, such as Qinghai Lake in China and other parts of Europe, and has since become a significant threat to the global poultry industry (5). In 1997, a male child lost his life as a result of being infected with H5N1 AIV (6), which marked the initial occurrence of cross-species transmission of AIV that led to the fatality. Then, other countries such as Japan and South Korea have continued to document individuals afflicted with H5N1 AIV (7). The considerable similarity observed in 1997 between avian and human influenza virus isolates indicated that H5-AIV was being transmitted from birds to humans (4). These occurrences serve to demonstrate that H5-AIV exhibits a broad range of transmission and a significant fatality rate, and due to its zoonotic nature, it presents a grave danger to the well-being of the general populace. Hence, the surveillance, prevention, and management of H5-AIV hold immense importance.

Generally, the conventional technique of isolating H5-AIV using chicken embryos is the most dependable and sensitive approach, but it is not feasible for large-scale detection and necessitates a BSL-3. On the other hand, the results of HA and hemagglutination inhibition (HI) tests are more straightforward to interpret, while this approach demands a substantial amount of time and effort. Real-time quantitative RT-PCR yields precise outcomes, but the equipment is costly and needs the expertise of skilled individuals (8). Consequently, there is an urgent requirement for a swift, portable, and precise method of determining results for the detection of H5-AIV.

In recent years, a plethora of isothermal amplification techniques have been employed for the detection of pathogenic infections, which include loop-mediated isothermal amplification (LAMP) (9), RAA10), nuclear acid sequence-based amplification11), and helicase-dependent amplification12). Besides, biosensors are seen as one of the directions of development for convenient, efficient, and low-cost detection of AIV, and several biosensor technologies have been developed for the detection of AIV (13, 14). Gene chip is a new high-throughput automated genetic testing technology. An oligonucleotide gene chip was used to simultaneously detect AIV, infectious bronchitis virus (IBV), Newcastle disease virus (NDV), and infectious laryngotracheitis virus (ILTV) with 100% specificity for AIV and comparable sensitivity to RT-PCR (15). RAA has emerged as a novel nucleic acid detection technique that boasts of rapid detection speed, exceptional specificity, and remarkable sensitivity. This technique has found extensive applications in the detection of viruses, bacteria, and parasites (16–18). The amplification of target gene fragments is primarily dependent on three essential enzymes: recombinase, single-stranded DNA-binding protein (SSB), and strand-substituted DNA polymerase. The formation of a protein-DNA complex through the interaction of recombinase and primers enables binding to homologous sequences within double-stranded DNA. After the primer identifies the homologous sequence, the single-stranded binding protein (SSB) separates the double-stranded template DNA. This allows for a strand exchange reaction to occur, facilitated by a strand-substituting DNA polymerase, resulting in the formation of a new complementary DNA strand. With a constant temperature of 37°C−42°C, the nucleic acid can be exponentially amplified within 30 min (19).

At present, there is a lack of information regarding the development of an RAA-based detection approach for H5-AIV. Given the severity of the avian influenza epidemic, it is imperative to facilitate prompt and precise detection of H5-AIV. To this end, we have developed an RT-RAA-LFD assay for H5-AIV, evaluated its specificity and sensitivity, and assessed its diagnostic performance in clinical settings through the use of clinical specimens.

## RESULTS

### Design and screening of H5-AIV RT-RAA primers and probe

Six pairs of forward and reverse primers were designed for use in both the RF-RT-RAA and RT-RAA-LFD assays; the primer and probe sequences were shown in Table 1. The amplification efficacy of those primers was evaluated through the use of the RF-RT-RAA assay, which was conducted at a temperature of 39°C for 20 min. During the process, the H5-AIV RNA standards served as the template. The results of RF-RT-RAA revealed that the employment of the F3/R1 led to the production of a curve with an early onset and a steep gradient. Consequently, the F3/R1 primer set was utilized in the subsequent experiments. To validate that the primer and probe sequences were specific to the H5-AIV, we compared them to four serotypes of AIV (H3-AIV, H7-AIV, H9-AIV, and H10-AIV), and the results are presented in Fig. 1. Most of the sites were found to be the same across different H5-AIV strains, with only a few sites exhibiting mutations, while distinct differences were observed in the sequences of H3-AIV, H7-AIV, H9-AIV, and H10-AIV compared to the sequences. Those results demonstrated that the primer and probe sequences employed were suitable for the detection of H5-AIV (Fig. 1).

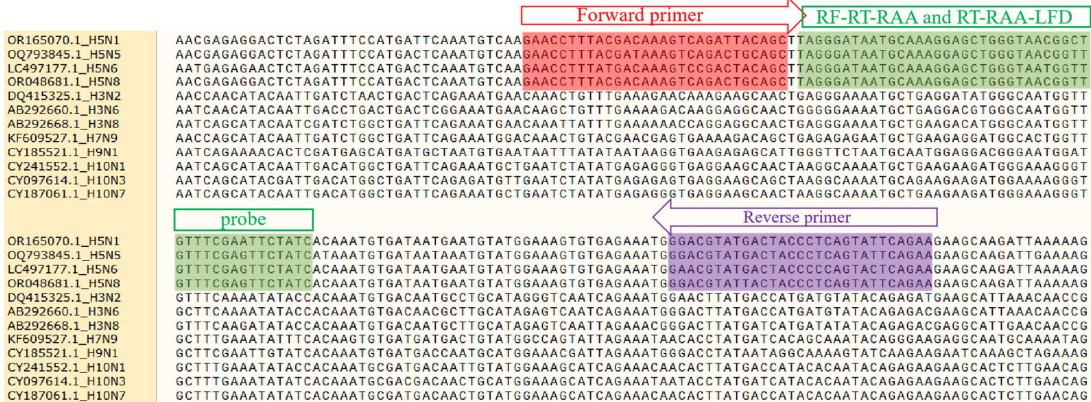

FIG 1   Results of sequence alignment of avian influenza virus subtype H5 sequence fragments with other similar avian influenza viruses.

TABLE 1   The sequence of primers and probes used in this study.

| Primers and probe | Sequences (5′-3′) |
| --- | --- |
| H5-PCR-F | GGAATATGGTAACTGCAACACCA |
| H5-PCR-R | AACTGAGTGTTCATTTTGTCAATG |
| RT-RAA F1 | AGAACTCTAGATTTCCATGACTCAAATGT |
| RT-RAA F2 | TTCCATGACTCAAATGTCAAGAACCTTTAT |
| RT-RAA F3 | GAACCTTTATGACAAAGTCCGACTACAGC |
| RT-RAA R1 | TTCTGAGTACTGGGGGGTAGTCATACGTTC |
| RT-RAA R2 | AATCTTGCTTCTTCTGAGTACTGGGGGTAG |
| RT-RAA R3 | CTTGCTTCTTCTGAGTACTGGGGGTAGT |
| H5-RAA Probe | TAGGGATAATGCAAAGGAGCTGGGTAATGG[6-FAM-dT][THF]G[BHQ1-dT]TTCGAGTTCTATC-3′C3 Spacer |
| H5-LFD Probe | FAM-TAGGGATAATGCAAAGGAGCTGGGTAATGGTT[THF]TTTCGAGTTCTATC -3′C3 Spacer |

## Specificity of the RF-RT-RAA assay

The specificity of the RF-RT-RAA assay was investigated by detecting RNA from different pathogens [H3-AIV, H7-AIV, H9-AIV, IBV, NDV, group A rotavirus (RVA), and duck astrovirus (DAstV)]. The results showed that only the H5-AIV exhibited a fluorescence amplification curve, while the other pathogens did not produce a positive signal (Fig. 2). Therefore, The RF-RT-RAA assay demonstrated a commendable specificity.

## Sensitivity of the RF-RT-RAA assay

The sensitivity of the RF-RT-RAA assay was evaluated by employing a series of RNA standards ranging from $10^3$ to $10^{-1}$ copies/µL as templates. The data demonstrated that a fluorescence amplification curve was evident when the concentration of RNA standard was above 0.1 copies/µL (Fig. 3). Therefore, the RF-RT-RAA assay for H5-AIV is capable of detecting as low as 1 copies/µL. As shown in Table 2, cRNA standards at concentrations of $10^3$ copies/µL and above were detected in eight replicates; whereas, when plasmid standards at $10^2$ copies/µL were used as templates, the H5-AIV RF-RT-RAA method showed positivity in seven out of eight replicates. When plasmid standards at $10^1$ copies/µL were used as templates, the H5-AIV RF-RT-RAA method showed positivity in five out of eight replicates and plasmid standards at $10^0$ copies/µL showed positivity in three out of eight replicates. Probit regression analysis was performed with the help of statistical software, and the LOD95 of RF-RT-RAA was 188 copies/µL.

## Establishment and optimization of the RT-RAA-LFD assay

The optimization of RT-RAA-LFD reaction temperature and time revealed that a reaction time of 12 min yielded no band at the test line (T) when using an RNA standards sample, while the band's intensity remained constant during the reaction periods of 15–18 min (Fig. 4). As for reaction temperature, the band was more prominent at a temperature of 37°C than at 39°C and 41°C (Fig. 5). Therefore, the optimal reaction time was selected as 15 min, and the optimal reaction temperature was 37°C.

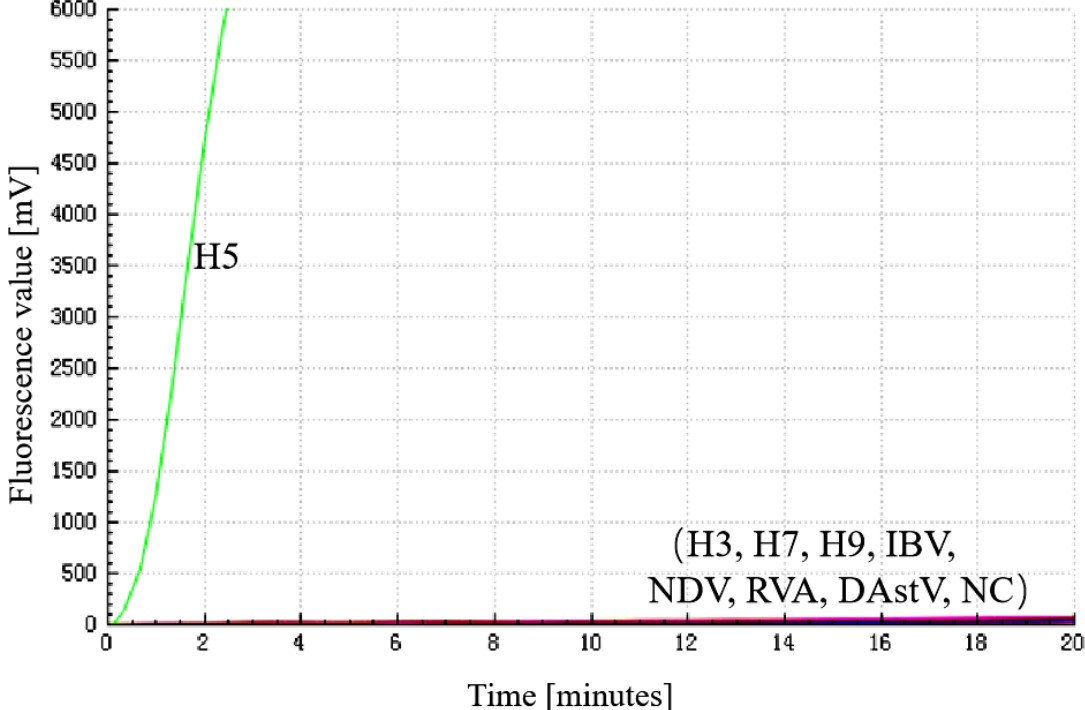

**FIG 2** Specificity of RF-RT-RAA assay for H5-AIV detection. NC was negative control; the specificity of the RF-RT-RAA method was investigated by using viral RNAs of H3-AIV, H7-AIV, H9-AIV, IBV, NDV, RVA, and DAstV as templates for detection; and the results indicated that none of the viral RNAs except H5-AIV was detected.

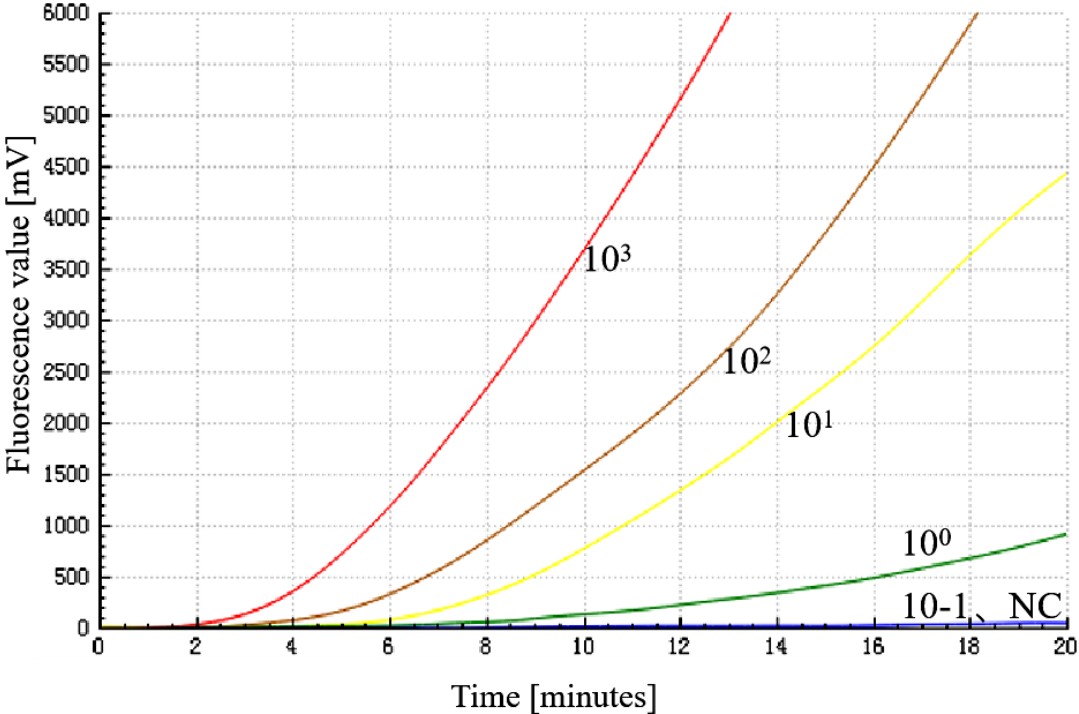

**FIG 3** Sensitivity of RF-RT-RAA assay for H5-AIV detection. NC was negative control, the sensitivity of the RF-RT-RAA assay was determined by using $10^3$–$10^{-1}$ copies/µL of the H5-AIV cRNA standard as templates, and the results revealed that the RF-RT-RAA assay was able to detect down to 1 copy/µL.

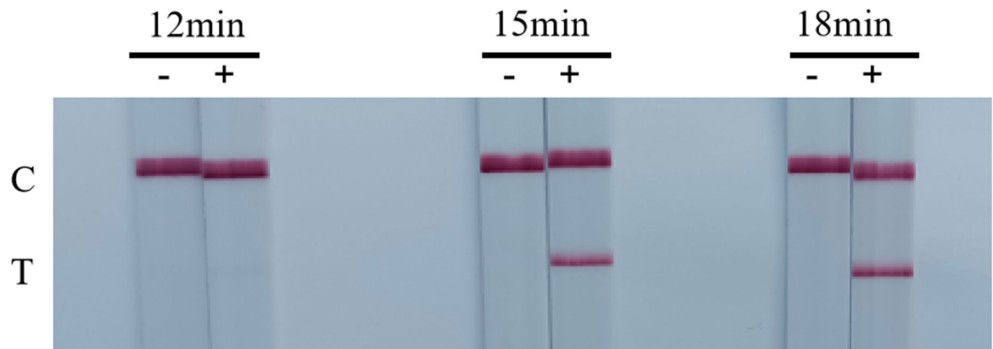

**FIG 4** Optimization of the RT-RAA-LFD reaction time. C was control line, T was test line, "−" was negative control, and "+" was positive control. The reaction time of RT-RAA-LFD assay was set at 12, 15, and 18 min, and the results showed that at 15 min, the positive control showed a clearly visible T-line and was no different from that at 18 min. The results revealed that the best reaction time was 15 min.

**TABLE 2** Assay date used for probit analysis to calculate the detection limit of H5 RF-RT-RAA and RT-RAA-LFD assay.

| Template concentration (copies/µL) | No. of positive sample/no. of samples tested | |
|---|---|---|
| | **RF-RT-RAA** | **RT-RAA-LFD** |
| $10^6$ | 8/8 | 8/8 |
| $10^5$ | 8/8 | 8/8 |
| $10^4$ | 8/8 | 8/8 |
| $10^3$ | 8/8 | 8/8 |
| $10^2$ | 7/8 | 6/8 |
| $10^1$ | 5/8 | 5/8 |
| $10^0$ | 3/8 | 2/8 |
| $10^{-1}$ | 0/8 | 0/8 |

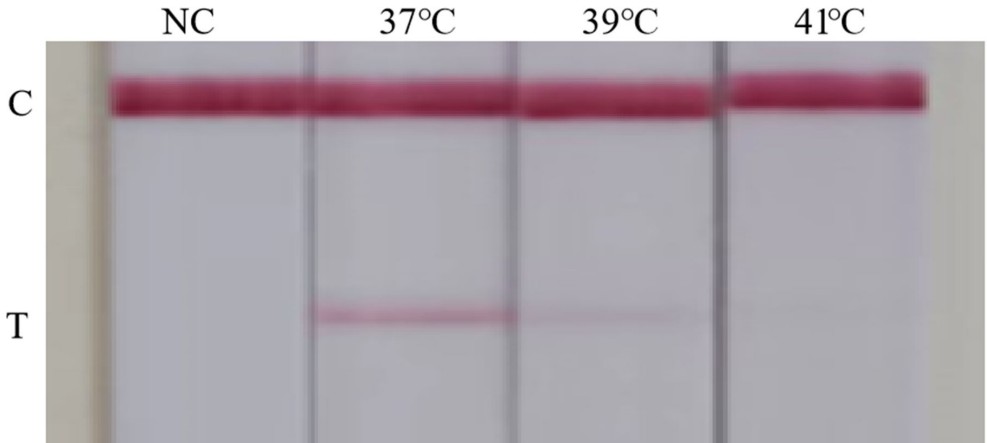

**FIG 5** Optimization of the RT-RAA-LFD reaction temperature. C was control line, T was test line, and NC was negative control. The reaction temperature for the RT-RAA-LFD assay was set at 37°C, 39°C and 41°C; the results showed that the T-line was brighter at a reaction temperature of 37°C than at the other two temperatures. The results revealed that the best temperature was 37°C.

## Specificity of the RT-RAA-LFD assay

To evaluate the specificity of the RT-RAA-LFD assay, RNAs of H3-AIV, H7-AIV, H9-AIV, IBV, NDV, RVA, and DAstV were used as templates for detection. At the same time, three H5-positive AIVs (all strain was in clade 2.3.4.4b) were used to evaluate the specificity of the RT-RAA-LFD. The result revealed that only the H5-AIV yielded visible test bands, whereas none of the other pathogens yielded negative signals, as shown in Fig. 6 and the H5-positive AIVs in clade 2.3.4.4b were also tested positive by RT-RAA-LFD assay. Therefore, the RT-RAA-LFD assay showed good specificity.

## Sensitivity of the RT-RAA-LFD assay

To evaluate the sensitivity of RT-RAA-LFD, various concentrations of RNA standards ($10^3$, $10^2$, $10^1$, $10^0$, $10^{-1}$ copies/μL) were utilized as templates and subsequently compared to RF-RT-RAA, RT-PCR, and real-time RT-PCR. The results revealed that the detection limit of RT-RAA-LFD was 1 copies/μL (Fig. 7a). The results of the RT-RAA-LFD test were found

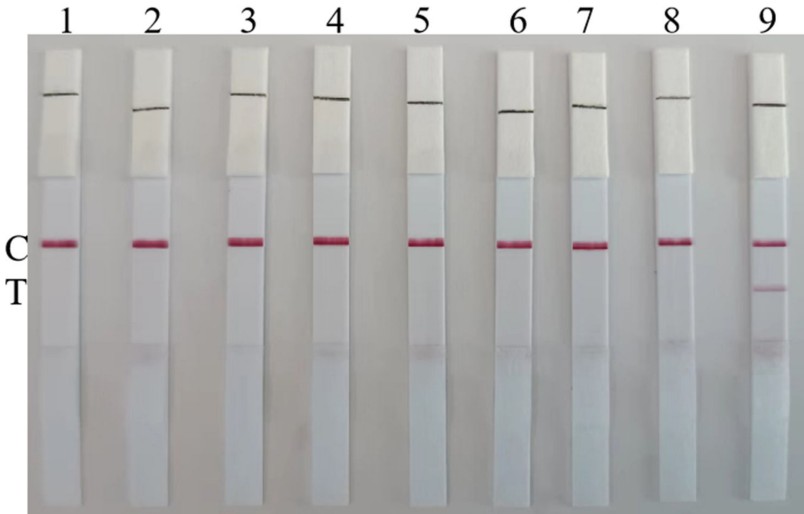

**FIG 6** Specificity of RT-RAA-LFD assay for H5-AIV detection. "C" was control line; "T" was test line; and 1–9 represents negative control, H3, H7, H9-AIV, IBV, NDV, RVA, DAstV, and H5-AIV, respectively. The results indicated that none of the viral RNAs except H5-AIV was detected.

to be consistent with those of the RF-RT-RAA test, exhibiting a sensitivity that was 100 times higher than that of RT-PCR (100 copies/µL) (Fig. 7b), while 10 times lower than that of real-time RT-PCR (0.1 copies/µL) (Fig. 7c). As shown in Table 2, cRNA standards at concentrations of $10^3$ copies/µL and above were detected in eight replicates; whereas, when plasmid standards at $10^2$ copies/µL were used as templates, the H5-AIV RT-RAA-LFD method showed positivity in six out of eight replicates. When plasmid standards at $10^1$ copies/µL were used as templates, the H5-AIV RT-RAA-LFD method showed positivity in five out of eight replicates and plasmid standards at $10^0$ copies/µL showed positivity in two out of eight replicates. Probit regression analysis was performed with the help of statistical software, and the LOD95 of RT-RAA-LFD was 406 copies/µL.

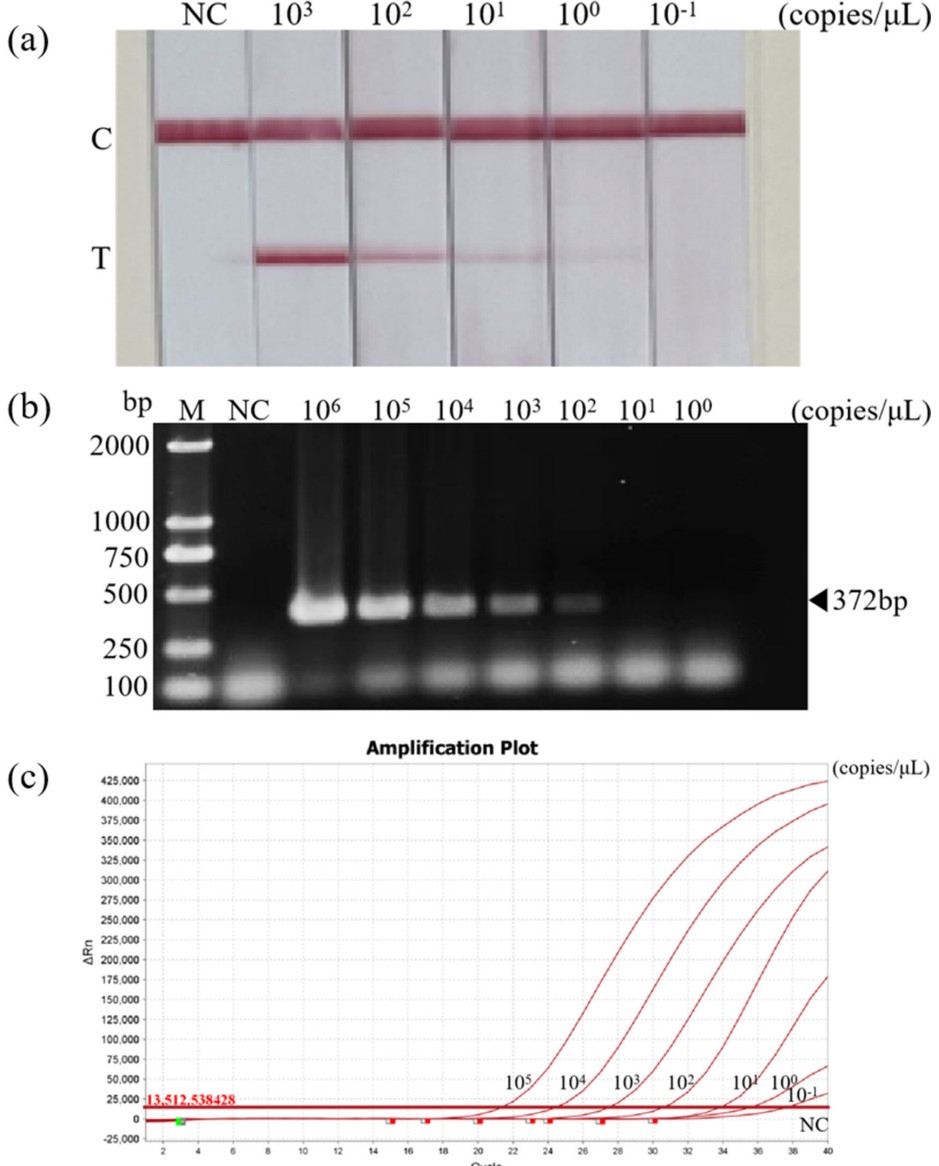

**FIG 7** Sensitivity of RT-RAA-LFD assay for H5-AIV detection. (a) Sensitivity of H5-AIV detection with RT-RAA-LFD at 37°C in 15 min. Serially diluted synthetic H5-AIV cRNA was used as a template. The top band is the control band, and the bottom band is the test band. (b) Sensitivity of H5-AIV detection with RT-PCR. (c) Sensitivity of H5-AIV detection with WOAH recommends real-time quantitative RT-PCR.

## Evaluation of the RT-RAA-LFD assay using clinical and experimental samples

To assess the clinical utility of RF-RT-RAA and RT-RAA-LFD for H5-AIV detection, 350 clinical swab samples and 26 experimental samples were tested, and the results were contrasted with those of virus isolation assay. Out of the total 350 clinical samples, 60 were found to be positive for H5-AIV through virus isolation. Upon detection through RF-RT-RAA, 55 samples were identified as positive for H5-AIV, whereas 295 were negative. Similarly, the result of RT-RAA-LFD showed 51 positive and 299 negative samples. The evaluation of field samples revealed that the kappa value between RF-RT-RAA and virus isolation assay was 0.948, while the level of agreement between RT-RAA-LFD and virus isolation assay was 0.904. Additionally, the RF-RT-RAA assay demonstrated a specificity and sensitivity of 100% and 91.67%, respectively, when compared with the virus isolation assay (Table 3). Similarly, the RT-RAA-LFD assay exhibited a specificity and sensitivity of 100% and 85%, respectively, in comparison to the virus isolation assay. As for 26 experimental samples, the RF-RT-RAA method detected 19 as positive, which was consistent with that of the virus isolation assay, while the RT-RAA-LFD method identified 18 of them as positive. The Kappa values for RF-RT-RAA and RT-RAA-LFD were 1 and 0.904, respectively, when compared to the viral isolation assay (Table 4).

## DISCUSSION

The ability to rapidly detect H5-AIV is a critical component in conducting epidemiological research and implementing measures to prevent and control its spread. Currently, the primary techniques for detecting H5-AIV include real-time RT-PCR, LAMP, and HA-HI. The SYBR Green real-time quantitative RT-PCR assay has a sensitivity of up to $10(0)$ $EID_{50}$/mL and 4.2 copies/µL for H5-AIV detection (20). However, this method necessitates costly equipment and skilled personnel to operate, which is time-consuming and unsuitable for prompt detection in manufacturing settings. The RT-LAMP technique employed for the detection of H5-AIV exhibits a sensitivity range of 100 to 1,000 copies per reaction, while the immuno-RT-LAMP assay for the identification of H5-AIV can detect as low as 16.2 $EID_{50}$/mL of virus in both normal and simulated viremia samples (21). However, the LAMP approach necessitates the use of multiple primer pairs, which can be considered a drawback. Although HI is widely regarded as the most reliable method for H5-AIV antibody detection, its implementation is hindered by the fact that it is a labor-intensive process that necessitates BSL-3 facilities and manual interpretation of results (22). As a result, it is not a viable option for primary laboratories and poultry farms.

**TABLE 3** Detection of H5-AIV in clinical samples using virus isolation, RF-RT-RAA and RT-RAA-LFD

|  |  | Virus isolation | | Total | Sensitivity (%) | Specificity (%) | Kappa |
|  |  | + | − |  |  |  |  |
|---|---|---|---|---|---|---|---|
| RF-RT-RAA | + | 55 | 0 | 55 | 91.67 | 100 | 0.948 |
|  | − | 5 | 290 | 295 |  |  |  |
|  | Total | 60 | 290 | 350 |  |  |  |
| RT-RAA-LFD | + | 51 | 0 | 51 | 85 | 100 | 0.904 |
|  | − | 9 | 290 | 299 |  |  |  |
|  | Total | 60 | 290 | 350 |  |  |  |

**TABLE 4** Detection of H5-AIV in experimental samples using virus isolation, RF-RT-RAA and RT-RAA-LFD

|  |  | Virus isolation | | Total | Sensitivity (%) | Specificity (%) | Kappa |
|  |  | + | − |  |  |  |  |
|---|---|---|---|---|---|---|---|
|  | + | 19 | 0 | 19 |  |  |  |
| RF-RT-RAA | − | 0 | 7 | 7 | 100 | 100 | 1 |
|  | Total | 19 | 7 | 26 |  |  |  |
|  | + | 18 | 0 | 18 |  |  |  |
| RT-RAA-LFD | − | 1 | 7 | 8 | 94.7 | 100 | 0.906 |
|  | Total | 19 | 7 | 26 |  |  |  |

The RAA assay, a novel isothermal amplification technique, has gained widespread popularity for its ability to amplify viral nucleic acids without the need for expensive equipment and complicated procedures. This technique can be performed at a temperature range of 37–42℃ in a time frame of 20–30 min (23). The reliability of the RAA assay is primarily contingent on the functionality of three key enzymes: a recombinase that utilizes pair-specific primers to target template DNA, a strand-replacement DNA polymerase that facilitates amplification and extension, and a single-stranded DNA-binding protein that binds to single-stranded DNA (16, 24, 25). Various methods can be employed to demonstrate the results of RAA, including fluorescence RAA, RAA in conjunction with LFD, and so on. LFD is widely used as a diagnostic tool for immune response, and its application with RAA assay enables the rapid visualization of results, which can be observed within 5–10 min (26). In recent years, the integration of RAA-LFD has facilitated the detection of various animal pathogens, including but not limited to porcine delta-coronavirus, 2019 novel coronavirus, and dengue virus (27–29).

In this study, we successfully developed two methods, namely, RF-RT-RAA and RT-RAA-LFD, for rapid detection of H5-AIV. To ensure the efficiency of primers and probes, we compared the HA gene of H5-AIV against multiple sequences and designed specific primers and probes based on conserved sequence regions. The results showed that the RF-RT-RAA and RT-RAA-LFD assays could be performed in just 20 and 15 min, respectively, indicating their rapidity and efficiency. Notably, both tests exhibited non-cross-reactivity with common avian viruses and were able to identify a minimum of 1 copy/µL. The clinical applicability of the RF-RT-RAA and RT-RAA-LFD assay was evaluated by utilizing 350 clinical samples and 26 experimental samples. When compared to the virus isolation assay, the RF-RT-RAA assay demonstrated a specificity and sensitivity of 100% and 91.67%, respectively, for detecting H5-AIV. Similarly, the RT-RAA-LFD assay exhibited a specificity and sensitivity of 100% and 85%, respectively. These findings indicated a high level of diagnostic agreement between RAA assays established in this study and real-time quantitative PCR. It has significant advantages over the method described in Table 5.

In conclusion, the RF-RT-RAA and RT-RAA-LFD assays established in this study have the potential to expedite the detection of H5-AIV. These procedures can be completed within 15 min at 37℃ and exhibit a remarkable sensitivity of 1 copy/µL. The assays are characterized by their high specificity, sensitivity, and user-friendliness. Moreover, the results of the reaction can be visually assessed, which is suitable for both laboratory testing and grassroots farm screening for H5-AIV.

## MATERIALS AND METHODS

### Biosafety statement and facility

The diagnosis and all experiments with live H5-AIV were conducted in biosafety level 3 facility in the China Animal Health and Epidemiology Center, which is approved for such use by the Ministry of Agriculture and Rural Affairs of the People's Republic of China.

### Virus strains

H3-AIV, H5-AIV, H7-AIV, H9-AIV, IBV, NDV, RVA, and DAstV were kept in the laboratory of the China Animal Health and Epidemiology Center.

### Collection of clinical and animal experiment samples

In the year 2021–2022, a total of 350 clinical swabs were collected randomly from live bird markets in Guangxi, Hunan, Henan, and Hebei provinces, China. The swabs were obtained from domestic chickens and ducks and stored at a temperature of −80℃. In addition, a group of 26 specific-pathogen-free (SPF) 6-week-aged chickens (Beijing Vital River Laboratory Animal Technology Co., Ltd, Beijing, China) were intranasally inoculated with H5-AIV at a dosage of $10^6$ $EID_{50}$. Post-mortem or 3 days post-inoculation

**TABLE 5** Results of different methods for detecting avian influenza virus

| Name | Method | Time (min) | Specitivity | Sensitivity | Clinical applications | Reference |
|---|---|---|---|---|---|---|
| H5-AIV | FICT | –[a] | No cross-reactivity with H1N1, H2N9, H3N2, H9N2, H5N3 LPAI, and influenza B viruses | 4 HAU/mL | High sensitivity for fecal samples and no positive results in cloacal samples | (30) |
| H9-AIV | Proposed immunosensor | 90 | No cross-reactivity with the remaining 15 AIVs | $10^{0.82}$ EID50/mL | Consistent with virus isolation in 98 clinical samples tested | (31) |
| H7N9 HA | SERS-LFIAS | 20 | No cross-reactivity with H5, H9-AIV, NDV, and IBDV | 0.0018 HAU | Consistent with virus isolation in 20 clinical samples tested | (13) |
| H5 and H9-AIV | LAMP | <30 | – | 100 to 1,000 RNA copies per reaction | – | (21) |
| H5-HA | Multiplex RT-qPCR | 90 | No cross-reactivity with human origin viruses, IBV, NDV, COV, and parainfluenza virus 1 | 50 copies/µL | Good agreement with gene sequencing results in 51 samples tested | (8) |
| H3, H5 H9-AIV | Multiplex RT-PCR | 65 | No cross-reactivity with H4N6 AIV, H6N1 AIV, IBV, IBDV, and a reovirus. | $10^1$ EID50/100 µL,$10^{-1}$ EID50/100 µL, and$10^0$ EID50/100 mL, respectively | Consistent with virus isolation in 392 clinical samples tested | (32) |
| H5, H7, H9-AIV | Oligonucleotide microarray | – | No cross-reactivity with any of the other avian respiratory viruses | 0.1 EID50/each reaction | Consistent with virus isolation results in 93 field samples tested | (33) |
| H5-AIV | RT-RAA | 20 | No cross-reactivity with H2N2, H2N3, H6N4, NDV, and IBV | $10^2$ copies/µL | The compliance rate of 420 clinical samples with the RT-qPCR was 99.5% | (34) |

[a]–, item is not mentioned in the reference.

experimental samples were collected from each chicken and detected by RF-RT-RAA and RT-RAA-LFD assays. To ensure consistency between the two assays, all samples were also inoculated into SPF chicken embryos for virus isolation, and HA-HI assays were performed as previously described to confirm the presence of H5-AIV in these samples.

## Primers and probe design

Under the design principle of RAA primers, a search was conducted on the NCBI database (https://www.ncbi.nlm.nih.gov/) for multiple sequence alignment of H5-AIV HA genomes. The target region for the detection of H5-AIV was selected based on the highly conserved region. For RF-RT-FAA, a series of primer sets were developed and subsequently evaluated for potential primer dimerization using the RF-RT-RAA kit (Qitian, Jiangsu, China). The RF-RT-FAA probe underwent FAM and BHQ1 modification, respectively, and 3′ end of the RF-RT-FAA probe was obstructed by phosphorylation and a dSpacer (THF) at a distance of ≥35 nt from its 5′-end. For RT-FAA-LFD, the reverse primer was subjected to biotinylation at its 5′ end, while the probe underwent fluorescein modification at the same position. All primers and probes were synthesized by Sangon Biotech (Shanghai) Co., Ltd. (Shanghai, China).

## Preparation of RNA standards used in this study

The viral RNA of H5-AIV was extracted in accordance with the instructions of the viral DNA/RNA nucleic acid extraction kit (Genfine Biotech, Beijing, China) and subsequently preserved at −80°C. The H5-AIV viral RNA was subjected to PCR amplification, and subsequently, the fragments were purified using a DNA purification kit (Thermo Fisher Scientific, China) and ligated to the pEASY-Blunt vector (TransGen Biotech, Beijing, China) to construct the recombinant plasmid *pEASY*-H5. Subsequently, the RNA standards were transcribed and purified by RiboMAX Large Scale RNA Production System-T7 kit (Promega, Beijing, China) using *pEASY*-H5 as

template DNA. The concentration of the RNA standards was determined using a Nano Drop nucleic acid quantitative analyzer, and the copy number was subsequently computed. The RNA standard underwent a continuous 10-fold serial dilution, which ranged from $10^3$ to $10^{-1}$ copies/µL.

## Establishment of RF-RT-RAA assay

The RF-RT-RAA reaction was successfully executed through the utilization of the RT-RAA kit (Zhongce, Hangzhou, China). The reaction buffer was premixed as follows: 25 µL A Buffer, 2 µL forward primer (10 µM), 2 µL reverse primer (10 µM), 0.6 µL probe (10 µM), 5 µL template RNA, and 12.9 µL ddH$_2$O. After thoroughly blending the aforementioned solution, 2.5 µL of B Buffer was introduced to the lids of each tube. Then, the RF-RT-RAA reaction mixture was mixed in a shaker mixer (Qitian, Jiangsu, China) and placed in thermostatic nucleic acid amplification assay (Qitian, Jiangsu, China) for 20 min at 37°C; the schematic of the RF-RT-RAA assay developed for rapid visual detection of H5-AIV is in Fig. 8.

## Establishment and optimization of RT-RAA-LFD assay

Similarly, the RT-RAA reaction was first premixed as following: 25 µL A buffer, 2 µL forward primer (10 µM), 2 µL reverse primer (10 µM), 0.6 µL probe (10 µM), 5 µL template, and 12.9 µL ddH$_2$O. Afterward, the mixture was subjected to amplification in a metal bath that had been thermostatically set to 37°C for a duration of 20 min. Subsequently, a total of 10 µL of the RT-RAA products were introduced into a 60 µL of dilution buffer and meticulously blended. Afterward, the LFD was inserted into the mixture, and the results were visualized within 3–5 min. In the event that both the control line (C) and test line (T) are visible simultaneously, the outcome shall be deemed positive. Conversely, if only the control line is observable, the result shall be considered negative; the schematic of the RT-RAA-LFD assay developed for rapid visual detection of H5-AIV is in Fig. 8 .

To enhance the accuracy of detection outcomes, we investigated the reaction conditions of RT-RAA-LFD assay. The reaction temperature was optimized by varying it at 37°C, 39°C, and 41°C while keeping the reaction time constant at 20 min. The optimization of reaction time was achieved by imposing a limit of 12, 15, and 18 min at 37°C. The optimal reaction conditions were identified based on the criteria of a clear detection line and the shortest time.

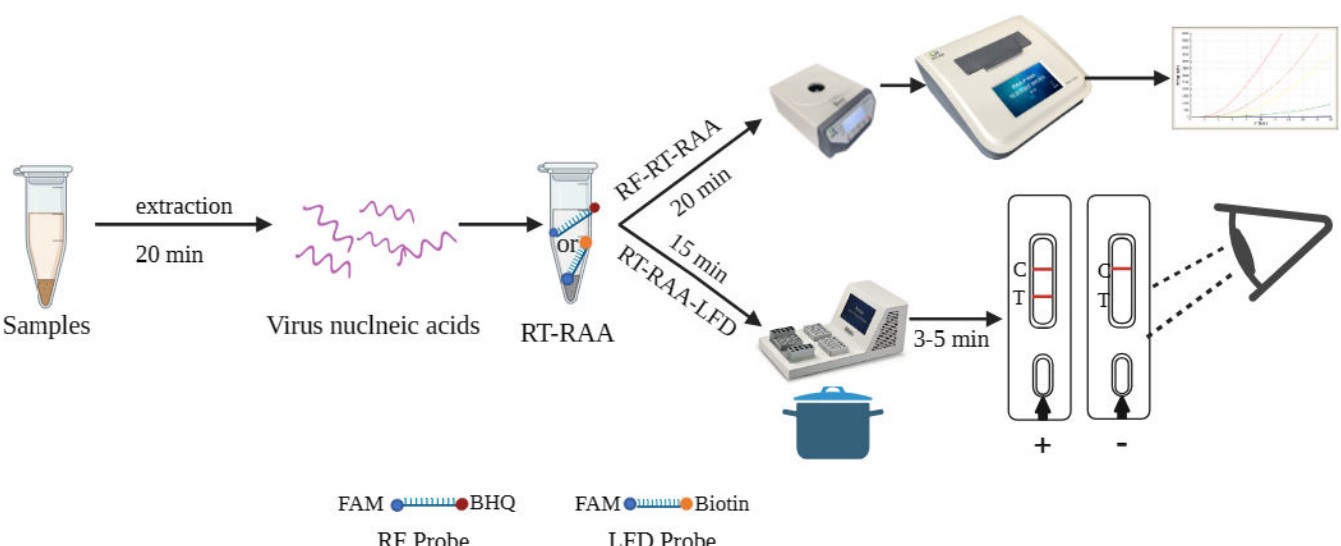

**FIG 8** Schematic of the RF-RT-RAA and RT-RAA-LFD assays developed for rapid visual detection of H5-AIV.

## Specificity and sensitivity testing of the RT-RAA-LFD assay

The specificity of the RT-RAA-LFD assay was tested using RNA of H3-AIV, H7-AIV, H9-AIV, IBV, NDV, RVA, and DAstV as templates and ddH$_2$O as negative control. The sensitivity of the RT-RAA-LFD assay was assessed using 10-fold serial dilutions of RNA standards ranging from $10^3$ to $10^{-1}$ copies/μL. The procedure was in accordance with the preceding text.

## Evaluation of the RT-RAA-LFD assay using clinical samples

Three hundred fifty clinical swab samples and 26 experimental samples were tested using the H5-AIV RF-RT-RAA and RT-RAA-LFD methods to evaluate the feasibility of two methods in clinical samples. Furthermore, the virus isolation assay was employed to examine the aforementioned samples, and a comparative analysis was conducted to evaluate the efficacy of the three assays.

## ACKNOWLEDGMENTS

This work was supported by grants from the National Key Research & Development Program (2022YFC2305101). All authors made substantial contributions to the work reported, H.L. and Y.L. designed the study conception. J.S., J.L., F.Z., and C.D. performed the experiments. W.J., X.Y., and X.Y. collected the clinical and animal experiment samples. J.S., J.L., F.Z., and C.D. analyzed the results. Y.L. and J.S. wrote the manuscript. Y.W. and H.L. revised the manuscript. The order of authors in this work was determined by workload. All authors agreed on the journal to which it should be submitted. All authors agree to be accountable for all aspects of the work.

## AUTHOR AFFILIATIONS

[1]China Animal Health and Epidemiology Center, Qingdao, China
[2]School of Life Science and Food Engineering, Hebei University of Engineering, Hebei, China
[3]College of Veterinary Medicine, Shandong Agricultural University, Shandong, China

## PRESENT ADDRESS

Yang Li, China Animal Health and Epidemiology Center, Qingdao, China

## AUTHOR ORCIDs

Yang Li  http://orcid.org/0009-0004-0577-9006
Jiajing Shang  http://orcid.org/0009-0005-7584-5889
HuaLei Liu  http://orcid.org/0000-0002-7680-1854

## FUNDING

| Funder | Grant(s) | Author(s) |
| --- | --- | --- |
| National key Research and Program | Nos. 2022YFC2305101 | Yang Li |

## AUTHOR CONTRIBUTIONS

Yang Li, Conceptualization, Writing – original draft | Jiajing Shang, Methodology, Writing – original draft | Yixin Wang, Writing – review and editing | Juan Luo, Methodology | Wenming Jiang, Resources | Xin Yin, Resources | Fuyou Zhang, Methodology | Chunran Deng, Methodology | Xiaohui Yu, Resources | HuaLei Liu, Writing – review and editing

## ETHICS APPROVAL

All experiments using animals were carried out in strict accordance with the recommendations in the Guide for the Care and Use of Laboratory Animals of the Ministry of Science and Technology of the People's Republic of China. All protocols were approved by the Animal Welfare Committee of the China Animal Health and Epidemiology Center (CAHEC).

## ADDITIONAL FILES

The following material is available online.

### Open Peer Review

**PEER REVIEW HISTORY (review-history.pdf).** An accounting of the reviewer comments and feedback.

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
