## [Reviewer comments · Microbiology Spectrum]

Microbiology Spectrum

Establishment of two assays based on reverse transcription recombinase-aided amplification technology for rapid detection of H5 subtype Avian Influenza Virus

Yang Li, Jia Shang, Yi Wang, Juan Luo, Wen Jiang, Xin Yin, Fu Zhang, Chun Deng, Xiao Yu, and Hua-Lei Liu

Corresponding Author(s): Yang Li, China Animal Health and Epidemiology Center

Review Timeline:

Submission Date:	May 30, 2023
Editorial Decision:	June 24, 2023
Revision Received:	August 9, 2023
Accepted:	August 10, 2023

Editor: Artem Rogovskyy

Reviewer(s): Disclosure of reviewer identity is with reference to reviewer comments included in decision letter(s). The following individuals involved in review of your submission have agreed to reveal their identity: Qingping Wu (Reviewer #2)

Transaction Report:

DOI: <https://doi.org/10.1128/spectrum.02186-23>

June 24, 2023

Dr. Yang Li
China Animal Health and Epidemiology Center
No.369, Nanjing Road, Shi Bei District, Qingdao
Qing Dao
China

Re: Spectrum02186-23 (Establishment of two assays based on reverse transcription recombinase-aided amplification technology for rapid detection of H5 subtype Avian Influenza Virus)

Dear Dr. Yang Li:

Link Not Available

Sincerely,

Artem Rogovskyy

Journals Department
Reviewer comments:

Reviewer #1 (Comments for the Author):

Reverse transcription recombinase-aided amplification technology is a well-developed detection method based on nucleic acid isothermal amplification. It has the advantages of no instrument requirements, very fast, and high detection sensitivity. In this study, a rapid detection method based on RT-RAA was developed for H5 subtype Avian Influenza Virus, and the detection results were displayed by fluorescence and lateral flow dipstick. The innovation of this work is mainly reflected in the application of RAA to a pathogen. Here are some comments:

1. The innovation of the work is reflected in the detection of a new virus, then a detailed introduction of primers and probes is

very necessary. What results show that the newly designed primers and probes are reliable? Although the specificity of the method was detected mainly by non-target viruses, is this method applicable to different variants of the target virus?

2. Were the primer sequences of conventional RT-PCR and real-time quantitative RT-PCR designed by the authors or was it a reference? It is well known that real-time RT-PCR is suitable for nucleic acid detection due to its high sensitivity. The authors compared the sensitivity differences between the new method and real-time RT-PCR, but why not compared the effects of the two methods when applied to samples? After all, they are both nucleic acid-based tests.

3. Detection is generally carried out for a type of pathogen, and specific type or variant strains can be identified by sequencing. So what is the significance of this subtype-specific testing method, would it not increase the cost of testing? In particular, what proportion of cases of avian influenza infection is caused by this subtype?

4. Line 39: " and field settings.". Nucleic acid extraction has always been a problem that needs to be solved in the application of this isothermal amplification rapid detection method. How did the authors overcome the nucleic acid extraction step?

5. Fig 4: Why does the RT-RAA-LFD method need to optimize the temperature, while the RF-RT-RAA method does not?
Fig 3: What concentration of RNA templates did the authors use for optimization? If the number of template copies is low, determine whether the reaction time needs to be extended.

6. LOD95 or LOD 90 is needed to determine the sensitivity of the method.

7. In the part of method application, what is the significance of using experimental samples?

Reviewer #2 (Comments for the Author):

Recommendation: The work is interesting, and it may be considered for publication in the journal after some major revisions. In this manuscript, Li and coworkers propose two analytical methods to successfully achieve rapid and sensitive detection of H5-AIV by the nucleic acid sequence of H5-AIV and combining RT-RAA technology. The proposed method is highly consistent with the traditional virus isolation method, and the test results of 376 practical samples show that. The proposed assay for H5-AIV also demonstrated high sensitivity and specificity. However, this manuscript should undergo some major revisions before being accepted by the journal *Microbiology Spectrum*.

1. There is no experimental principle or flow chart of the method in this manuscript, especially for RT-RAA-LFD, which should be introduced to summarize the proposed method more intuitively and briefly.

2. What are the highlights of this method according to the author?

3. Supplement the title of "Introduction" in the manuscript, and list in this part the existing emerging detection technologies in addition to traditional methods for detecting AIV.

4. In the section of the "Discussion", the first half of the paragraph involves a lot of background introduction, which should be in the introduction rather than the discussion.

5. The manuscript should be supplemented by recently published results of different methods for detecting AIV and other avian influenza viruses, presented in tabular form.

6. In the section of the "Results", line 110, the author proposed that the optimal reaction was 39°C, but 37°C was selected for the subsequent experiment. In line 127, the RT-RAA-LFD showed that not produce color in the T-line within 12min, while in line 293, it was proposed that color development could occur within 3-5min. Please give a reasonable explanation for this.

7. In this manuscript, the author mentions that the optimum temperature for RAA reaction is between 37-42°C, and in the optimization of reaction temperature, 37°C is obviously better than the other two temperatures. Please give the possible reasons for such a large difference in results under the optimum reaction temperature.

Staff Comments:

Preparing Revision Guidelines

- Point-by-point responses to the issues raised by the reviewers in a file named "Response to Reviewers," NOT IN YOUR COVER LETTER.
- Upload a compare copy of the manuscript (without figures) as a "Marked-Up Manuscript" file.

- Each figure must be uploaded as a separate file, and any multipanel figures must be assembled into one file.
- Manuscript: A .DOC version of the revised manuscript
- Figures: Editable, high-resolution, individual figure files are required at revision, TIFF or EPS files are preferred

Please return the manuscript within 60 days; if you cannot complete the modification within this time period, please contact me. If you do not wish to modify the manuscript and prefer to submit it to another journal, please notify me of your decision immediately so that the manuscript may be formally withdrawn from consideration by Microbiology Spectrum.

Dear Editor and Reviewer:

Thank you for your kind letter of “Spectrum02186-23 Decision Letter” on June 24, 2023. we would like to appreciate the editor and reviewer for the positive and constructive comments and suggestions. We have studied comments carefully and have substantially revised our manuscript which we hope meet with approval. Based on your comments and requests, we have made modification on the original manuscript. In revision notes, the line numbers refer to the PDF vision of the revised manuscript.

Responses to reviewer #1 comments:

Thanks for your constructive suggestions, and we have highlighted the changes in yellow according to reviewer #1 comments in the revised manuscript.

Q1. The innovation of the work is reflected in the detection of a new virus, then a detailed introduction of primers and probes is very necessary. What results show that the newly designed primers and probes are reliable? Although the specificity of the method was detected mainly by non-target viruses, is this method applicable to different variants of the target virus?

Re: We are thankful to the reviewer’s insightful comments. To ensure the specificity and sensitivity of primers and probes, we compared and analyzed sequences of different H5 subtypes and similar AIV, and the results are shown in Fig. 1. According to Fig. 1, most of the loci in the sequences of the different H5 subtypes of AIVs are conservative, apart from few of the mutated loci. At the same time, there were marked variations in the sequences of the other three subtypes of viruses (H3, H7, H9 and H10-AIV) in comparison to the H5-AIV. To ensure the accuracy of this outcome, we conducted an experiment to verify our findings, and the results showed that different H5 subtypes of AIV (H5N1, H5N3, H5N6, and H5N8) could be detected using this assay.

Q2. Were the primer sequences of conventional RT-PCR and real-time quantitative RT-PCR designed by the authors or was it a reference? It is well known that real-time RT-PCR is suitable for nucleic acid detection due to its high sensitivity. The authors compared the sensitivity differences between the new method and real-time RT-PCR, but why not compared the effects of the two methods when applied to samples? After all, they are both nucleic acid-based tests.

Re: We are thankful to the reviewer’s insightful comments. Firstly, the primer sequences of

conventional RT-PCR and real-time quantitative RT-PCR were referenced from Chinese national standard (GB/T 18936—2020). We have supplemented a "Source" column to Table 1. We chose to use the virus isolation method rather than real-time RT-PCR when testing the effect of the new method on the samples as both are the "gold standard" for detecting H5-AIV in international standards, and the real-time RT-PCR results were in agreement with those of the virus isolation method when we tested clinical samples. Furthermore, owing to our capability of isolating HP-AIV, we opted to compare the virus isolation method with the new method for H5-AIV.

Q3. Detection is generally carried out for a type of pathogen, and specific type or variant strains can be identified by sequencing. So what is the significance of this subtype-specific testing method, would it not increase the cost of testing? In particular, what proportion of cases of avian influenza infection is caused by this subtype?

Re: We are thankful to the reviewer's insightful comments. The development of a method to detect H5-AIV was necessitated by the requirement to safeguard human health and the poultry farming industry from the considerable menace posed by H5-AIV. Traditional detection methods call for more laborious and time-consuming processes; for example, PCR must be followed by agarose gel electrophoresis to sequence the fragments that correspond to the H5-AIV, although this approach is more reliable, the entire cycle may take up to two days, making it a lengthy procedure, and the delays in detecting the virus can be fatal in case of an outbreak on farms. To ensure the procedure is exclusively successful for the H5-AIV, primers and probes tailored specifically for these viruses were employed, thus minimizing the time of sequencing steps and guaranteeing accurate results.

In regards to the percentage of H5-AIV in the total number of avian influenza infections, we would like to provide some clarification. The World Organisation for Animal Health (WOAH) only legally requires notification of subtypes H5 and H7-AIV. As the remaining subtypes have not been added to the WOAH database, we are unable to ascertain the proportion of cases caused by these subtypes in the total number of cases caused by AIV. In 2022, the H5N1-AIV outbreaks resulted in the slaughter or culling of more than 131 million poultry in over 60 countries around the world. WOAH reported that since 2005, 389 poultry have been killed or destroyed by various H5-AIV. Early detection of the H5-AIV is essential in order to minimize losses in the poultry industry.

Q4. Line 39: "and field settings.". Nucleic acid extraction has always been a problem that needs to be solved in the application of this isothermal amplification rapid detection method.

How did the authors overcome the nucleic acid extraction step?

Re: We are thankful to the reviewer's insightful comments. Indeed, we have yet to develop a kit that effectively combines nucleic acid extraction and virus detection in a timely fashion. In this manuscript, in the nucleic acid extraction section, we used a commercially available centrifugal column-type nucleic acid extraction kit, which requires a palm centrifuge to carry out the process, and the whole extraction process was controlled within 20 min. As we move forward, we will collaborate with our associates to address this issue.

Q5. Fig 4: Why does the RT-RAA-LFD method need to optimize the temperature, while the RF-RT-RAA method does not? Fig 3: What concentration of RNA templates did the authors use for optimization? If the number of template copies is low, determine whether the reaction time needs to be extended.

Re: We are thankful to the reviewer's insightful comments. For Fig. 4, we had already implemented a rapid RT-RAA assay for DAsV-3 before this study. Through the process of optimising the reaction temperature, it was discovered that the three temperatures of 37 °C, 39 °C, and 41 °C had minimal effect on the results of RF-RT-RAA amplification, and thus, after following the kit instructions, 39 °C was chosen as the optimal reaction temperature. Consequently, for this experiment, the reaction temperature was not optimized but rather 39°C was adopted as the temperature for the reaction. We observed that prolonged reaction times are capable of detecting decreased template concentrations, however, the false-positive rate increases when the test paper's reaction time is extended to identify a lower template concentration.

Q6. LOD95 or LOD 90 is needed to determine the sensitivity of the method.

Re: We are thankful to the reviewer's insightful comments. As you can see in Table 2, in the detection of H5-AIV using both methods, cRNA standards at 10^3 copies/ μ L and above were detected in both methods in eight repetitions of the test: whereas, when plasmid standards at 10^2 copies/ μ L were used as templates, 7 out of 8 repetitions showed positivity by the RF-RT-RAA and 7 out of 8 repetitions by the RT-RAA-LFD methods; and when plasmid standards at 10 copies/ μ L were used as templates, 5 out of 8 repetitions showed positivity by the RF-RT-RAA method, and 5 out of 8 repetitions showed positivity by the RT-RAA-LFD method; and when plasmid standards at 1 copies/ μ L was used as the template, the RF-RT-RAA method showed positivity for 3 times in 8 repetitions, and the RT-RAA-LFD method showed positivity for 2 times in 8 repetitions. Probit regression analysis was performed with the help of statistical software, and the LOD95 of the RF-RT-RAA and RT-RAA-LFD assays were 188 copies/reaction and 406 copies/reaction,

respectively.

Q7. In the part of method application, what is the significance of using experimental samples?

Re: We are thankful to the reviewer's insightful comments. To ensure the applicability of the two methods in both laboratory and farm settings, we employed experimental samples in addition to clinical samples in the application section of the method to evaluate if there were any significant discrepancies between the two methods in detecting known laboratory samples and unknown clinical samples.

Reviewer #2 (Comments for the Author):

The work is interesting, and it may be considered for publication in the journal after some major revisions. In this manuscript, Li and coworkers propose two analytical methods to successfully achieve rapid and sensitive detection of H5-AIV by the nucleic acid sequence of H5-AIV and combining RT-RAA technology. The proposed method is highly consistent with the traditional virus isolation method, and the test results of 376 practical samples show that. The proposed assay for H5-AIV also demonstrated high sensitivity and specificity. However, this manuscript should undergo some major revisions before being accepted by the journal Microbiology Spectrum.

Re: Many thanks. We appreciate the reviewer's carefulness and insightful comments. We have highlighted **changed contents** in Purple.

Q1. There is no experimental principle or flow chart of the method in this manuscript, especially for RT-RAA-LFD, which should be introduced to summarize the proposed method more intuitively and briefly.

Re: We are thankful to the reviewer's insightful comments. We have supplemented the RF-RT-RAA and RT-RAA-LFD assay flow charts to the manuscript for your inspection.

Q2. What are the highlights of this method according to the author?

Re: We are thankful to the reviewer's insightful comments. Utilizing specifically designed primers and probes for H5-AIV, this study employed rapid real-time fluorescence and lateral flow dipstick based on the sensitive and specific RAA technique instead of conventional H5-AIV detection methods that require complex instrumentation. This method was found to be more efficient than the commonly used real-time RT-PCR and RT-PCR methods, as it reduced the detection time from 1-2 hours to 30 minutes. What's more, the sensitivity of this method was better than RT-PCR and

comparable to real-time RT-PCR. These methods provide a convenient way to detect on-site without the requirement of large instruments and expensive reagents, making them suitable for field testing or locations with limited technical resources.

Q3. Supplement the title of "Introduction" in the manuscript, and list in this part the existing emerging detection technologies in addition to traditional methods for detecting AIV.

Re: We are thankful to the reviewer's insightful comments. I am sorry for our carelessness. We have supplemented the title "Introduction" in the manuscript.

Q4. In the section of the "Discussion", the first half of the paragraph involves a lot of background introduction, which should be in the introduction rather than the discussion.

Re: We are thankful to the reviewer's insightful comments. We have removed this part and added it to the introduction and marked it with highlight.

Add content in 55-56: Deleted the original line 55 sentence of "AIV primarily affects domestic poultry such as chickens, ducks, and turkeys," and replace it with "AIV can infect a multitude of animal species, including but not limited to chickens, turkeys, pigs, and humans".

Add content in 68-73: Deleted the original line 68-69 sentence of "H5-AIV was first identified in geese in Guangdong province 68 of China in 1996", and replaced it with "In 1996, the highly pathogenic H5N1 AIV was initially identified in a group of diseased geese in Guangdong, China, and was designated as A/Goose/Guangdong/1/96, GS/GD/96. The virus initially emerged in China, but it eventually spread to the Middle East and Europe. Additionally, there were reports of the same virus in wild birds in various places, such as Qinghai Lake in China and other parts of Europe and has since become a significant threat to the global poultry industry.

Add content in 73-77: Before the original 70 line, added the sentences as following: "In 1997, a male child lost his life as a result of being infected with H5N1 AIV, which marked the initial occurrence of cross-species transmission of AIV that led to fatality. Then, other countries such as Japan and South Korea have continued to document individuals afflicted with H5N1 AIV."

Add content in 79: Added the sentences "These occurrences serve to demonstrate that before original line 72.

Add content in 95-101: Added the sentences as following: "nuclear acid sequence-based amplification (NASBA), and helicase-dependent amplification (HAD). Besides, biosensors are seen as one of the directions of development for convenient, efficient and low-cost detection of AIV, and several biosensor technologies have been developed for the detection of AIV. Gene chip

is a new high-throughput automated genetic testing technology. An oligonucleotide gene chip was used to simultaneously detect AIV, IBV, NDV and ILTV with 100% specificity for AIV and comparable sensitivity to RT-PCR” before original line 87.

Add content in 129-135: Added the sentences as following: “To validate that the primer and probe sequences were specific to the H5-AIV, we compared them to four serotypes of AIV (H3-AIV, H7-AIV, H9-AIV, and H10-AIV), and the results are presented in Fig. 1. Most of the sites were found to be the same across different H5-AIV strains, with only a few sites exhibiting mutations, while distinct differences were observed in the sequences of H3-AIV, H7-AIV, H9-AIV, and H10-AIV compared to the sequences. Those results demonstrated that the primer and probe sequences employed were suitable for the detection of H5-AIV. (Fig. 1).”

Q5. The manuscript should be supplemented by recently published results of different methods for detecting AIV and other avian influenza viruses, presented in tabular form.

Re: We are thankful to the reviewer’s insightful comments. We have supplemented different methods for detecting AIV at the end of the manuscript with the title of “The results of different methods of detecting avian influenza viruses”.

Q6. In the section of the "Results", line 110, the author proposed that the optimal reaction was 39 °C, but 37 °C was selected for the subsequent experiment. In line 127, the RT-RAA-LFD showed that not produce color in the T-line within 12 min, while in line 293, it was proposed that color development could occur within 3-5 min. Please give a reasonable explanation for this.

Re: We are thankful to the reviewer’s insightful comments. For Fig4, we had established a Rapid RT-RAA Assay for DAsV-3. For line 110, 39 °C was set as the optimum reaction temperature for RF-RT-RAA because our established “Establishment of a Rapid RT-RAA Assay for DAsV-3 did not differ significantly when performing reaction temperature optimisation, as the kit recommended temperature of 39 °C was chosen for this experiment. When performing RT-RAA-LFD, we reviewed a large body of literature and most of them were optimised to ensure accurate results while saving assay time.

For line 127, 12 min is the RT-RAA amplification time and 3-5 min in line 293 is the time to read the results, i.e. when performing reaction time optimisation, the positive template is amplified for 12 min and then diluted and the result is negative when the test strip is placed in it and observed within 3-5 min. The reason for setting the time to observe the results within 3-5 min is to avoid false positive results.

Q7. In this manuscript, the author mentions that the optimum temperature for RAA reaction is between 37-42 °C, and in the optimization of reaction temperature, 37 °C is obviously better than the other two temperatures. Please give the possible reasons for such a large difference in results under the optimum reaction temperature.

Re: We are thankful to the reviewer's insightful comments. 37-42 °C is the temperature range in which recombinase and polymerase can amplify efficiently in the RAA reaction, the most suitable reaction temperature in a reaction depends on one's experimental conditions, because the binding efficiency of our primers and enzymes affects the amplification of the target, in addition, the different reaction systems also affect the selection of the optimal reaction temperature to a certain extent.

Minor concerns:

Line 114: “a RAA-based” was changed to “an RAA-Based”.

Line 125: “39 °C for a duration of 20 minutes” was changed to “39 °C for 20 minutes.”.

Line 126: “RNA standards was served as the template” was changed to “RNA standards served as the template”.

Line 139: “while the other pathogens did not produce positive 139 signal” was changed to “while the other pathogens did not produce a positive 139 signal”.

Line 143: “RNA standards ranging 10^3 to 10^{-1} copies/ μ L” was changed to “RNA standards ranging from 10^3 to 10^{-1} copies/ μ L”.

Line 157: “when using RNA standards sample, while the.....” was changed to “when using an RNA standards sample, while the.....”.

Line 162: “Specificity of the RT-RAA-LFD assay. In order to evaluate the specificity of the.....” was changed to “Specificity of the RT-RAA-LFD assay. To evaluate the specificity of the.....”.

Line 165: “was in clade 2.3.4.4b) were used to evaluated.....” was changed to “was in clade 2.3.4.4b) were used to evaluate.....”.

Line 167: “pathogens yielded negative signals, as show in Figure 6 a” was changed to “pathogens yielded negative signals, as shown in Figure 6 a”.

Line 168: “Therefore, RT-RAA-LFD assay showed good specificity” was changed to “Therefore, the RT-RAA-LFD assay showed good specificity”.

Line 170: “Sensitivity of the RT-RAA-LFD assay. In order to evaluate the sensitivity.....” was changed to “Sensitivity of the RT-RAA-LFD assay. To evaluate the sensitivity.....”.

Line 186: “To assess clinical utility of RF-RT-RAA.....” was changed to “To assess the clinical utility of RF-RT-RAA.....”.

Line 196: “compared with virus isolation assay (Table 3).” was changed to “compared with the virus isolation assay (Table 3).”.

Line 198: “virus isolation assay. As for 26 experimental samples” was changed to “the virus isolation assay. As for 26 experimental samples”.

Line 199: “which was consistent with that of virus isolation assay, while” was changed to “which was consistent with that of the virus isolation assay, while”.

Line 201: “....., when compared to viral isolation assay (Table 4).” was changed to “....., when compared to the viral isolation assay (Table 4).”.

Line 206: “Currently, the primary techniques for detecting H5-AIV includes real-time RT-PCR” was changed to “Currently, the primary techniques for detecting H5-AIV include real-time RT-PCR”.

Line 229: “LFD is widely used as a 229 diagnostic tool for immune response, which its application with.....” was changed to “LFD is widely used as a 229 diagnostic tool for immune response, and its application with.....”.

Line 231: “which can be observed within a span of 5-10 minutes (26)” was changed to “which can be observed within 5-10 minutes (26)”.

Line 233: “including but not limited to porcine delta coronavirus, 2019.....” was changed to “including but not limited to porcine delta-coronavirus, 2019.....”.

Line 237: “we compared the HA gene of H5-AIV against multiple sequences, and” was changed to “we compared the HA gene of H5-AIV against multiple sequences and”.

Line 253: “Moreover, the results of the reaction can be visually assessed, which are suitable.....” was changed to “Moreover, the results of the reaction can be visually assessed, which is suitable”.

Line 279: “In accordance with the design principle of RAA.....” was changed to “Under the design principle of RAA.....”.

Line 304: “The reaction buffer was premixed as following: 25 µL.....” was changed to “The reaction buffer was premixed as follows: 25 µL.....”.

Line 304: “10-fold serial dilutions of RNA standards ranging $10^3\sim 10^{-1}$ copies/µL.” was changed to “10-fold serial dilutions of RNA standards ranging from $10^3\sim 10^{-1}$ copies/µL.”.

Line 350: “H5-PCR-F” was changed to “GGAATATGGTAACTGCAACACCA ”; H5-PCR-R was changed to “AACTGAGTGTTTCATTTTGTCAATG”; H5-LFD Probe was changed to

“ FAM-TAGGGATAATGCAAAGGAGCTGGGTAATGGTT/THF/TTTCGAGTTCTATC -3C3

Spacer.

Line 397: “EFERENCES” was changed to “REFERENCES”

August 10, 2023

Dr. Yang Li
China Animal Health and Epidemiology Center
No.369, Nanjing Road, Shi Bei District, Qingdao
Qing Dao
China

Re: Spectrum02186-23R1 (Establishment of two assays based on reverse transcription recombinase-aided amplification technology for rapid detection of H5 subtype Avian Influenza Virus)

Dear Dr. Yang Li:

Your manuscript has been accepted, and I am forwarding it to the ASM Journals Department for publication. You will be notified when your proofs are ready to be viewed.

Sincerely,

Artem Rogovskyy
Editor, Microbiology Spectrum
